# A Python diagnostics package for evaluation of MJO-Teleconnections in S2S forecast systems

- Cristiana Stan<sup>1</sup>, Saisri Kollapaneni<sup>1</sup>, Andrea M. Jenney<sup>2</sup>, Jiabao Wang<sup>3</sup>, Zheng Wu<sup>4</sup>, Cheng Zheng<sup>5,6</sup>,
- Hyemi Kim<sup>7</sup>, Chaim I. Garfinkel<sup>8</sup>, and Ayush Singh<sup>2</sup>

- Department of Atmospheric, Oceanic and Earth Sciences, George Mason University, Fairfax, Virginia, USA
- <sup>2</sup>College of Earth, Ocean, and Atmospheric Sciences, Oregon State University, Corvallis, USA
- <sup>3</sup>Center for Western Weather and Water Extremes, Scripps Institute of Oceanography, University of California, San Diego,
- USA
- 4Department of Atmospheric and Environmental Sciences, University at Albany, SUNY, Albany, USA
- <sup>5</sup>Lamont-Doherty Earth Observatory, Columbia University, USA
- <sup>6</sup>School of Marine and Atmospheric Sciences, Stony Brook University, USA
- <sup>7</sup>Department of Science Education, Ewha Womans University, Seoul, Republic of Korea
- 8The Fredy and Nadine Hermann Institute of Earth Sciences, Hebrew University of Jerusalem, Jerusalem, Israel

16

25

- Correspondence to: Cristiana Stan (cstan@gmu.edu)
- **Abstract.** The MJO-Teleconnections diagnostics package is an open-source Python software package that provides process-
- level evaluation of MJO-Teleconnections predicted by subseasonal-to-seasonal (S2S) forecast systems. The package provides
- in-depth process-level evaluation of both tropospheric and stratospheric pathways defining the atmospheric teleconnections
- from the tropics to extratropics on S2S time scales. The analyses include comparison of a forecast model with a default
- verification data set or user-provided verification data. The package consists of a user-friendly graphic user interface (GUI),
- which allows the package to be applied to both operational and research models. This approach allows for efficient data
- management and reproducibility of analysis.

# 1 Introduction

- S2S forecasting is an activity that bridges the gap between medium-range weather forecast (lead time 1-2 weeks) and the
- seasonal forecast (lead time 1-3 months). In this time range, the main sources of predictability transition from the memory of
- initial conditions to the boundary conditions forcing. As a result, other sources of predictability are tapped into for enhancing
- the forecast skill for weeks 3-4 lead time. One of the sources of predictability, especially for boreal winters, has been identified
- as the extratropical response to the Madden Julian Oscillation (MJO; Madden and Julian 1971, 1972) activity in the tropical
- atmosphere. The MJO signal reaches the Northern Hemisphere (NH) midlatitudes via barotropic Rossby waves (Jin and
- Hoskins 1995, Wang and Xie 1996) channelled through a waveguide in the upper troposphere and/or through stratosphere-
- troposphere coupling. Teleconnections following the 'tropospheric pathway' have a global impact, with the strongest influence

on the Pacific North American sector's surface weather (Higgins et al. 2000, Cassou 2008, Lin et al. 2009, Zhou et al. 2012, Riddle et al. 2013, Johnson et al. 2014). Teleconnections following the 'stratospheric pathway' influence the surface weather of the North Atlantic and Europe. They can constructively interact with the teleconnections through the upper troposphere, amplifying the response to MJO forcing. This results in a stronger and more long-lasting effect (Schwartz and Garfinkel 2017, Green and Furtado 2019).

The MJO is the dominant large-scale atmospheric circulation pattern of the intraseasonal variability (30-90 days) in the tropics. A typical event manifests as a 'pulse' of cloud and rainfall that moves eastward at about 4-8 m/s and recurs every 40 to 60 days (Madden and Julian 1971, 1972). The MJO primary peak season is boreal winter when the strongest signals are located south of the equator; the secondary peak season is boreal summer when the strongest signals are located north of the equator (Zhang 2005). The amplitude and phase of MJO events can be described using the Real-time Multivariate MJO (RMM) index (Wheeler and Hendon, 2004). The life cycle of an MJO event is divided into eight phases described by the location of increased cloudiness and rainfall (active convection) as follows: in phase 1, it begins over the western Indian Ocean; in phase 2, it moves eastward to the central Indian Ocean; in phase 3, it reaches the eastern Indian Ocean and Maritime Continent; in phase 4, it spreads over the Maritime Continent; in phase 5, moves over the far-western Pacific; in phase 6, spreads over the western hemisphere and Africa, completing the cycle.

The Rossby waves form in response to perturbations induced by moist diabatic processes associated with tropical convection (Teng and Branstator 2019), which oftentimes aggregates into the MJO. The waves are generated by the so called Rossby wave sources (Sardeshmukh and Hoskins 1988) or basic state vorticity gradients. The MJO heating leads to horizontal divergence of the wind in the upper troposphere and changes in the rotational wind. The wave activity emanating from the source propagates eastward and poleward. While the waveguide is modulated by the position of jet streams (Enomoto et al. 2003), it can also be influenced by the anomalous cyclonic and anticyclonic upper level circulations induced by the waves (Zheng et al. 2018).

The waves with zonal wavenumber-1 and wavenumber-2 play the most important role in the stratosphere-troposphere coupling as they transit the tropopause and reach the polar stratosphere (Charney and Drazin 1961, Weinberger et al. 2022). The heat and momentum fluxes transported by these waves can be absorbed within the stratospheric polar vortex, modifying the strength of the vortex (Chen and Robinson 1992, Limpasuvan et al. 2004, Polvani and Waugh 2004, White et al. 2019, Weinberg et al 2022). Perturbations in the vortex are often associated with changes in the large-scale circulation patterns at the surface in the following weeks (Baldwin and Dunkerton 2001, Polvani and Kushner 2002, Polvani and Waugh 2004, White et al. 2019, Baldwin et al. 2021).

The impact of the MJO on the extratropics is stronger during boreal winter and manifests as regional modulations of circulation, temperature and precipitation (Stan et al. 2017) across the entire NH (Yoneyama and Zhang 2020). The response of the extratropics to MJO forcing establishes in a two-week time scale (Jin and Hoskins, 1995) and depends on the MJO phases described by the location of active and suppressed convection. Stan et al. 2017 provides a review of mechanisms underpinning the tropical-extratropical teleconnections.

Given the broad impact of the MJO on global weather/climate systems, acting as a major source of global S2S predictability, this paper introduces a new Python package that consists of metrics and diagnostics for evaluation of the MJO and processes driving the MJO teleconnections in forecast data. The scientific basis of diagnostics included in the package have been documented in literature. The diagnostics have been applied to the forecast systems in the S2S database (Stan et al. 2022) and the prototypes of the NOAA UFS global coupled model (Zheng et al. 2024, Garfinkel et al. 2024, Wang et al. 2025).

The objective of the paper is to guide users in how to apply the package to their forecast data, understand the strength and weaknesses of a forecast system in predicting the mechanisms driving the MJO teleconnections compared to their observed characteristics, and to provide a limited deterministic evaluation of the forecast skill. Additionally, the package provides a tool for evaluation of the MJO forecast skill. Due to the delayed response of the extratropics to MJO forcing, evaluation of MJO teleconnections by forecast systems is conducted with respect to the presence of MJO events in initial conditions, which allows the usage of reanalysis/observation based products for event description.

The package consists of a GUI and a collection of modular evaluation tools, all written in Python v3.9.16 and its associated scientific libraries. The package can be applied to any forecast dataset prepared in the specified format. The basic concept of the diagnostics package is similar to other community-contributed metrics packages such as the PCMDI Metric Package (PMP; Gleckler et al. 2008, 2016; Lee et al. 2024), the Toolkit for Extreme Climate Analysis (TECA), the international Land Model Benchmarking Tool (ILAMB; Collier et al. 2018), the International Ocean Model Benchmarking (IOMB) package operating under the umbrella of the Coordinated Model Evaluation Capabilities (CMEC), the Process-oriented diagnostics (PODs) coordinated by the Model Diagnostics Task Force (MDTF; Neelin et al. 2023), the Climate Variability and Diagnostics Package (CDVP; Phillips et al. 2014, Maher et al. 2024), the Atmosphere Model Working Group (AMWG) Diagnostics Framework (ADF), and others. The major difference between these diagnostics packages and the MJO-Teleconnections diagnostics package is that the latter is tailored for S2S forecast data, which typically extend to no more than 46 days (Vitart et al. 2017). The other packages apply to multi-century climate simulations.

Figure 1: Window-based structure of GUI (a) Selection of function mode. (b) Window collecting user input for running diagnostics.

# 2 Components of the diagnostic package

108109

111112

115116

122123

2.1 The graphical user interface The user interface provides two primary functions: selecting and running diagnostics, and displaying existing results (Fig. 1a). The interface is built using PyOt5, a popular Python framework for creating GUIs. The GUI features a window-based design organized into a sequence of menus with two sections. The static section provides help text, while the dynamic section contains input containers and buttons for user interaction. (Fig. 1b). By providing quick information, the help text reduces the need to consult external documentation. Users can navigate back and forth between windows, allowing for a flexible workflow. The diagnostics can be applied to a single forecast dataset, which can be compared against a default dataset or user-provided verification data. The default datasets are the ECMWF Reanalysis (ERA) Interim (Dee et al. 2011) for wind, geopotential and 2-meter temperature and the Integrated Multi-satellitE Retrievals for GPM, (IMERG; Huffman 2014) for precipitation. The user inputs collected through the GUI are saved as dictionaries into a YAML configuration file. Then, each diagnostic reads relevant entries from this configuration. This approach confers the package flexibility and extensibility. The design allows for easy addition of new diagnostics. Users can choose from various diagnostics options and the interface allows running a single diagnostic, a subset, or all available diagnostics. After the computation of the diagnostics is completed, the interface opens a new window for displaying the results of all computed diagnostics (Fig. 2). Figures and in some cases data files can then be saved locally. The GUI offers additional options such as computing or using pre-computed model anomalies, using the default RMM index or a user provided index, and specifying forecast details such as time period, length, and number of ensembles. Help text is provided to guide users on file formats and names as well as variable names and units (Table A1 in the Appendix A provides the list of accepted variable names and units).

Figure 2: GUI windows illustrating the computational stage of the diagnostic.

- Windows displaying results contain comprehensive explanations of the main features emphasized by the diagnostic. This format allows users with different abilities to use the diagnostics, reducing the need to consult external documentation. Having the result of each diagnostic available in a stand alone window allows users to compare diagnostics results simultaneously.
- The GUI has built-in features that prevent users from inputting invalid data. For example, no field can have a null value and users are prompted with additional information about the missing information.
  - It is also possible to select and run diagnostics without the GUI. In non-GUI mode, users can directly specify diagnostic settings with a YAML configuration file, which the package reads to run the desired diagnostics. The same capabilities are available in non-GUI mode as in the GUI version, ensuring consistency across different usage modes and allowing for automation, batch processing, and integration into larger workflows without the need for manual interaction with the graphical interface.

# 2.2 Diagnostics

The MJO-Teleconnections codebase is designed to be modular and each diagnostics set is self-contained. They share common tools such as horizontal interpolation, computation of climatology and anomalies. Each diagnostic tool has its own main python script that is invoked by GUI and reads the user input from the configuration YAML file. The output from each diagnostic including figures and tables are organized in a user defined directory and can be displayed in GUI.

All diagnostics require data in the network Common Data Form (netCDF) format using CF Metadata conventions (Eaton et al. 2024). Forecast data must be organized in one experiment per file and per ensemble member for diagnostics requiring calculations for individual ensemble members. If the forecast is compared with ERA Interim (ERAI) and IMERG and the horizontal resolution of the forecast data is higher than the verification data, each diagnostic interpolates the forecast data to the ERAI grid (512 longitude grid points ordered from 0° to 360°, 256 latitude grid points ordered from 90N to 90S) and IMERG grid (480 longitude grid points ordered from 0° to 360°, 241 latitude grid points ordered from 90S to 90N), respectively. The stored direction of forecast data's latitude can be either decreasing or increasing. If the horizontal resolution of the forecast data is coarser than that of verification data, the interpolation is to the grid of forecast data. The code ensures that regriding is always from the high to low resolution grid. The regriding algorithm is Python Spherical Harmonic Transform Module, pysharm 1.0.9. (Whitaker, 2020), which is built on the collection of FORTRAN programs SPHEREPACK (Adams and Swartztrauber 1999). Verification data can also be user specified and in that case, the grid must match the grid of the forecast data. The package does not have the capability to directly compare two sets of forecast data. In all diagnostics, the MJO phases are defined based on the MJO phase characterizing the initial date of the forecast. Users have the option to use the RMM index based on ERAI or provide new index data.

The meteorological parameters used by the package include: geopotential or geopotential height at 500 and 100 hPa, (Z500 and Z100), temperature at 500 and 100 hPa (T500 and T100), zonal and meridional components of the wind at 850 hPa (U850 and V850), zonal wind at 10 hPa (U10), meridional component of the wind at 500 hPa (V500), zonal component of the wind at 200 hPa (U200), 2-meter temperature (T2m), surface precipitation rate (PREC), and outgoing longwave radiation (OLR) at the top of the atmosphere. Table 1 in the Appendix provides the accepted names of variables and their units. All diagnostics with the exception of one require data at daily frequency. The forecast data must consist of a minimum of 35 days.

In all diagnostics, the MJO teleconnections are evaluated during boreal winter (November through March). For diagnostics that include statistical significance, a bootstrap method using 1000 samples is used for its computation.

# 2.2.1 STRIPES Index

The Remote Influence of Periodic EventS (STRIPES) index was introduced by Jenney et al. (2019) to characterize the regional impact of aggregated MJO phases, each being assumed to last approximately 5 to 7 days. The STRIPES index measures the MJO teleconnections as the covariability between the MJO activity and regional fluctuations of meteorological parameters. It takes into account both the magnitude and consistency of the MJO's influence across multiple events. The STRIPES index takes values between 1 and +∞. It reaches high values when two conditions are met: first, the meteorological parameter shows strong correlation with the MJO activity, and second, this relationship remains consistent across multiple MJO events.

Figure 3: STRIPES index for geopotential height at 500 hPa (left) and precipitation (right) computed using observations (ERAI and IMERG) and forecasts (Model) at forecast lead time weeks 2-3. Bottom panels show the difference between model and observations.

In the package, the STRIPES index is computed for Z500 and PREC for the global domain. The result is displayed for the verification data, forecast data, and the difference between the two corresponding to three forecast leads: weeks 1-2, 2-3, and 3-4. Figure 3 shows an example of the STRIPES index for each variable in week 2-3 of the forecast. The STRIPES index for Z500 in the NH shows that in ERAI the strongest impact of MJO manifests over regions along the Atlantic and Pacific storm tracks and Europe. In the Southern Hemisphere (SH), the response to MJO forcing is a zonally elongated belt around 60S. The forecast captures the approximate location of the response centers in both hemispheres. However, the magnitude of the response is weaker in the NH and stronger in the SH as shown by the difference plot. The STRIPES index applied to IMERG shows that the response of extratropical precipitation to MJO forcing is localized over the same regions as the circulation response. The Model forecasts a weaker than observed amplitude of the response in both hemispheres.

This index identifies regions where forecast models capture or miss the regions where the MJO has a significant and predictable impact on the large-scale circulation and precipitation as well as the strength of the impact. The caveat for the SH is that during boreal winter MJO teleconnections to this hemisphere are weak.

STRIPES index depends on the number of MJO events used for its calculation. The shorter the analyzed period, the larger the sensitivity of the index. If users want to compare the STRIPES index computed using two sets of forecasts, the number of

MJO events or analyzed periods must be the same for the two sets. Figure 4 shows the STRIPES index computed using two different periods of ERAI: 8 and 18 years, respectively. A comparison of the STRIPES index for the two periods shows that

Figure 4: STRIPES index for geopotential height at 500 hPa weeks 2-3 after MJO events computed using ERAI between 2011-2018 (left) and 2002-2019 (right).

calculation based on the shorter period yields in maximum values of the STRIPES index larger than in the calculation based on a longer period.

#### 2.2.2 Pattern Correlation and Relative Amplitude

The two extratropical regions in the NH with the strongest MJO influence are the Pacific North America and North Atlantic. This can be seen in Fig.3 and in many other studies (for a complete review see Stan et al. 2017). These studies also show that MJO teleconnections are not uniform across the MJO phases. The influence on the large-scale Pacific North American (PNA) region is dominated by the MJO convective activity in the tropical Indian Ocean and western Pacific (Mori and Watanabe 2008), also known as phases 2-3 and 6-7 when using the RMM index. The influence of MJO on the large-scale circulation over the North Atlantic and Eurasia is robust 10-15 days after the occurrence of MJO phase 3 and 7 (Cassou 2008, Lin et al. 2009).

The ability of the forecast model to capture these relationships can be evaluated using the pattern correlation coefficient (pattern CC) and relative amplitude metrics applied to Z500 in each region. The diagnostics for the PNA region are constructed over a domain between  $20^{\circ}-80^{\circ}$ N,  $120^{\circ}$ E $-60^{\circ}$ W (Wang et al. 2020) and for Euro-Atlantic over the area between  $20^{\circ}-80^{\circ}$ N,  $60^{\circ}$ W $-90^{\circ}$ E. The pattern CC is between the Z500 daily anomalies in the forecasts and observations. The relative amplitude is defined as the standard deviation of Z500 daily anomalies in the model divided by that in observations. Figure 5 shows the metrics for the PNA region when MJO events in phases 2-3 and 6-7 are present in the initial conditions of the forecasts. In this example, the prediction of MJO teleconnection pattern is skillful (pattern CC > 0.6) up to two weeks regardless of the MJO phase. In the first week, the amplitude of MJO teleconnection in the forecast data is close to ERAI (relative amplitude  $\sim$  1). As the lead time increases, the Model alternates between underestimating (relative amplitude < 1) and overestimating (relative amplitude > 1) the amplitude of the MJO teleconnections to the PNA region for both MJO phases. At shorter leads (weeks 2-3) the

amplitude is largely underestimated following the MJO phases 6-7 whereas at longer leads (weeks 3-4) the overestimation of amplitude is more pronounced for the MJO phases 2-3. Before launching the calculation of the metrics, users have the option to select the computation of composites of Z500 daily anomalies over the NH used in the calculation of metrics (Fig. 6). In the window displaying the figures corresponding to pattern CC and relative amplitude users have the option to download a text file of the values in coma-separated values (csv) format.

Figure 5: Pattern cc (model vs. observations) and relative amplitude of Z500 daily anomalies over the PNA region (20°-80°N, 120°E-60°W) as a function of forecast lead days for the MJO phases 2-3 (blue) and 6-7 (red). The shading indicates the 95% confidence level determined by a bootstrap test of 1000 samples. The lower (upper) boundary represents the 2.5th (97.5th) percentile of the bootstrapping distribution.

Figure 6 shows that in ERAI, the Z500 composites after the MJO phases 2-3 resemble the negative PNA pattern and the composites after the MJO phases 6-7 resemble the positive PNA pattern. The composites based on the forecast data capture the Z500 anomaly pattern well in the first two weeks of the forecast after both the MJO phases 2-3 and 6-7 and the accuracy decreases at longer leads. One particular aspect to notice is the model failure to capture the transition of Z500 anomalies associated with the MJO phases 2-3 to the opposite MJO phases (6-7) occurring in week 3 in ERAI. The pattern of the MJO teleconnections in the forecast tends to agree better with ERAI over the North Pacific than over North America indicating some model deficiencies in predicting the patterns over the land than over the ocean.

Figure 6: Weekly averaged composites of Z500 daily anomalies at lead time week 1 (a-d), week 2 (e-h), week 3 (i-l) and week 4 (m-p) for observations (ERAI) and forecast (Model). In the left (right) two columns forecasts have MJO events in phases 2 and 3 (6 and 7) present in the initial conditions. Numbers in the right upper corners show the pattern CC between model and observations over the PNA region ((20°-80°N, 120°E-60°W).

These diagnostics help identify model deficiencies related to the pattern and amplitude of specific MJO's phases impact on the large-scale circulation over the PNA and Euro-Atlantic regions.

# 2.2.3 Frequency distribution of zonal wind at 10 hPa

As an initial step in evaluating the stratospheric pathway of teleconnections, this diagnostic is designed to determine the response of the polar vortex to MJO forcing. The polar vortex experiences two opposite extreme conditions: sudden stratospheric warming (SSWs) and strong polar vortex (SPV) events. SSWs occur when dynamic forcings disrupt the stratospheric circulation resulting in predominantly westward-flowing winds across much of the polar stratosphere. SPVs are associated with strong westerly zonal mean zonal winds caused by anomalously weak planetary wave activity. These extreme events can have impacts that extend to the Earth's surface. The zonal mean zonal wind at 60°N and 10 hPa is used for characterizing the extreme events in the polar vortex (Charlton and Polvani 2007, Charlton-Perez and Polvani 2011, Smith et al. 2018, Oehrlein 2020; Baldwin et al 2021). The diagnostic evaluates the distribution of zonal mean zonal winds at 60°N and 10 hPa averaged over weeks 1-2 and 3-5 after MJO events in phases 1-2 and 5-6 are present in the initial conditions of the forecast (Stan et al. 2022, Garfinkel et al. 2024). Figure 7 shows an example of the zonal mean zonal wind histograms averaged over weeks 1-2 of the forecast.

Figure 7: Histograms of daily values of zonal mean zonal wind at 10 hPa 60°N and (U1060) in November-March for forecast (Model) and observations (ERAI) for weeks 1-2 following the MJO phases 1-2 (blue) and phases 5-6 (yellow). The solid blue lines indicate mean values of U1060 during the respective MJO phases. The dashed blue and yellow lines indicate the 5th and 95th percentile of U1060 during the MJO phases 1-2 and 5-6, respectively.

This diagnostic helps identify the strength of the polar vortex following the MJO activity present in the initial conditions of the forecast. The diagnostic helps assess the influence of MJO on the stratosphere in models. The stratosphere-troposphere coupling diagnostics following the MJO activity described in the next section provide a deeper analysis of processes in the model that need to be improved.

#### 2.2.4 Stratosphere-troposphere coupling

The stratosphere-troposphere coupling is evaluated using two diagnostics: i) the meridionally averaged (40° - 80°N) meridional heat flux anomaly associated with quasi-stationary planetary waves with wavenumber 1 and 2 at 500 hPa and ii) the geopotential height anomaly at 100 hPa averaged over the polar cap (55° - 90°N). Both diagnostics are computed for MJO phases 1-8 and forecast leads week 1 through 5. An enhanced positive heat flux anomaly 2-3 weeks after MJO in phase 5 is typically associated with the upward propagation of heat fluxes entering the stratosphere followed by weakening of the polar vortex. The downward propagation from the stratosphere into the troposphere is shown by the polar cap geopotential height anomaly. Positive polar cap height anomalies in weeks 3-5 following MJO in phase 6 tend to indicate the negative phase of Northern Annular Mode (NAM) and its downward propagation. Figure 8 shows an example of the stratosphere-troposphere coupling diagnostics.

Figure 8: Meridionally averaged heat flux anomalies at 500 hPa (top) and 100 hPa geopotential height anomalies averaged over the polar cap (70° - 90°N, 0 - 360) for observations (ERAI) and forecast (Model) in weeks 1-5 following MJO phases 1-8 during November-March.

In this case, the Model is not able to maintain the strength of the positive heat flux associated with the MJO phase 5 beyond week 2. In fact, the Model reverses the sign of the heat flux anomaly at longer forecast leads, suggesting an opposite response of the polar vortex, which is confirmed by the 100 hPa geopotential height anomalies averaged over the polar cap seen in weeks 3-5 following the MJO phase 5.

This diagnostic helps identify the upward troposphere to stratosphere coupling using the planetary wave activity flux and the downward stratosphere to troposphere coupling using the NAM anomaly following the MJO activity. The diagnostic helps assess the stratospheric pathway of MJO teleconnections.

# 2.2.5 Extratropical cyclone activity

The package offers two diagnostics tools for evaluating the complex relationship between the MJO and extratropical cyclone activity. One diagnostic computes the composites of eddy kinetic energy (EKE) at 850 hPa for the MJO phases at week 3 - 4 forecast leads. The EKE is constructed from winds filtered to retain the synoptic-scale variability with periods between 1.2 and 6 days. For this reason, the data required for this diagnostic is recommended to be specified at 6-hourly frequency. The diagnostic can also work with daily mean or 24-hourly data, however, the EKE calculation is sensitive to this aspect, especially if the sample size is relatively short. The calculation of EKE also needs to be done for each ensemble member before calculating the ensemble mean. The composites are computed for the model and a verification dataset including their statistical significance. Different phases of the MJO are considered because different phases can lead to shifts in preferred storm tracks for extratropical cyclone activity. This can affect which regions are more likely to experience cyclone activity during specific MJO phases. Certain phases of the MJO are associated with increased intensity of extratropical cyclone activity in specific regions. To measure the agreement between the storm tracks activity predicted by the model and the verification data set, the pattern correlation is computed. An example of the composites is shown in Fig. 9. A similar analysis is conducted for Z500 to represent the large-scale circulation variability associated with the MJO. The second diagnostic is designed to measure the correspondence between the changes in large-scale circulation induced by MJO and downstream effects that can be more or less favourable for extratropical cyclone formation and intensification. The impact of model errors in the MJO-induced largescale circulation on the errors in the extratropical cyclone activity can be determined by plotting the pattern correlation of eddy kinetic energy at 850 hPa versus the pattern correlation of Z500, for various MJO phases and forecast leads. An example of this diagnostic for the North Atlantic and North Pacific storm tracks regions is shown in Fig. 10.

Figure 10: Pattern correlation of week 3-4 composites of EKE850 (y-axis) and Z500 (x-axis) between ERAI and Model for the North Atlantic (20°-80°N, 90°W-30°E) and the North Pacific and North America (20°-80°N, 120°E-90°W). Different colors represent different MJO phases.

This diagnostic helps identify model deficiencies in simulating storm track activity modulated by the MJO. The MJO's influence on extratropical cyclones is often associated with synoptic-scale extreme precipitation and temperature events (e.g., Kunkel et al. 2012; Ma and Chang 2017).

#### 2.2.6 MJO

Due to the time lag between the MJO activity and extra-tropical response, the diagnostics are built using the MJO events present in the initial conditions of the forecasts. However, how the forecast models maintain these events is important for the correct prediction of the MJO teleconnections. For example, the ability of models to maintain the coherence of MJO convection across the Maritime Continent is important for a correct prediction of MJO phases 2 and 3. The propagation speed of MJO events can also influence the MJO teleconnections. For instance, rapidly propagating MJO events result in weak westerlies across the North Atlantic high latitudes (Yadav and Straus, 2017; Yadav et al. 2019). Slowly propagating MJO events can weaken the polar vortex and downstream contribute to a relatively stronger impact over the North Atlantic and Eurasia (Yadav et al. 2024). The diagnostics package calculates the RMM1 and RMM2 indices, following the method in Gottschalck et al. (2010) using U850 and U200 from ERAI and NOAA OLR (Liebman and Smith 1996). These are then used to compute the bivariate anomaly correlation (ACC), root mean square error (RMSE), and amplitude (AERR) and phase error (PERR) between the forecast and observations. The eastward propagation of the MJO pattern is evaluated using Hovmoller diagrams of the daily anomalies for OLR and zonal wind at 850 hPa averaged over 15°S-15°N for active MJO events in the initial conditions of the forecast. Examples of these metrics are shown in Fig. 11. The definitions of ACC, RMSE, AERR and PERR are adopted from Rashid et al. (2011):

$$ACC(\tau) = \frac{\sum_{t=1}^{N} [a_1(t)b_1(t,\tau) + a_2(t)b_2(t,\tau)]}{\sqrt{\sum_{t=1}^{N} [a_1(t)^2 + a_2(t)^2]} \sqrt{\sum_{t=1}^{N} [b_1(t,\tau)^2 + b_2(t,\tau)^2]}}$$

$$RMSE(\tau) = \sqrt{\frac{1}{N} \sum_{t=1}^{N} ([a_1(t) - b_1(t, \tau)]^2 + [a_2(t) - b_2(t, \tau)]^2)}$$

$$AERR(\tau) = \frac{1}{N} \sum_{t=1}^{N} \left( \sqrt{b_1(t,\tau)^2 + b_2(t,\tau)^2} - \sqrt{a_1(t)^2 + a_2(t)^2} \right)$$

$$PERR(\tau) = \frac{1}{N} \sum_{t=1}^{N} tan^{-1} \left( \frac{a_1(t)b_2(t,\tau) - a_2(t)b_1(t,\tau)}{a_1(t)b_1(t,\tau) + a_2(t)b_2(t,\tau)} \right)$$

where  $a_1$  and  $a_2$  are RMM1 and RMM2 in observations,  $b_1$  and  $b_2$  are RMM1 and RMM2 in forecast data, t is for initialization time with a lead time of  $\tau$  days, and N is the total number of predictions. As noted by Rashid et al. (2011) this is not equivalent to the difference between an average phase of the forecasts and observations and positive *PERR* values indicate the MJO in the forecasts leads the event in observations.

The AERR and PERR metrics show that the Model predicts MJO events with weaker amplitude and faster phase speed than in ERAI. The ACC shows that the forecast skill drops below 0.5 after  $\sim$ 27 days at about the same time when RMSE = 1.4. The Hovmoller diagram shows that in the model convective activity takes a longer time to propagate across the Maritime continent.

Figure 11: Phase (PERR) and amplitude (AERR) errors (top left), bivariate anomaly correlation (ACC) and root mean square error (RMSE) (top right; the gray solid horizontal line indicates an ACC of 0.5 and RMSE of 1.25). Longitude-time composites of OLR (W/m2; shading) and U850 (contour; interval 0.3 m/s) anomalies averaged over 15°S-15°N for active MJO events in observations (ERAI) and forecast (Model). The results are for events initialized during MJO phases 2 and 3. The vertical lines indicate 120°E (approximately the center of the Maritime Continent). A 5-day moving average is applied.

This diagnostic helps identify the errors in the amplitude, phase speed of the MJO predicted by the model. It also shows the model's ability to predict the propagation of the MJO across the Maritime Continent and the coupling between the circulation and convective activity associated with the MJO events.

### 2.2.7 Surface air temperature

The impact of the MJO on the surface air temperature is evaluated in the composites of T2m anomalies for forecasts that have MJO events in phases 3 and 7 present in the initial conditions. The phases are chosen because they are known to have the strongest teleconnections to temperature patterns in the NH. Composite maps are constructed for forecast leads ranging from week 1 to week 5 and the observed counterparts. The composite maps are displayed as stereographic projections of the NH and include the statistical significance. To quantify the accuracy of the forecasts, pattern correlation between the forecast data and verification data is computed for each composite map. This metric provides a measure of how well the pattern of predicted

T2m matches the observed patterns. Because of the typical 15-day time lag between MJO activity and observed temperature changes in the NH, the first two weeks provide information on model errors due to local processes affecting the temperature anomaly patterns. If the MJO diagnostic indicates a slower (faster) propagation of the MJO than in observations the errors in the T2m pattern can also be attributed to the timing of the MJO impact on the existing temperature anomalies. The strength of the MJO events (e.g., measured by the Relative amplitude diagnostic) can also affect the length of the lag and the effect of MJO teleconnection to the region. Figure 12 shows an example of T2m composites in week 3 after MJO events in phase 3 are present in the initial conditions of the forecasts.

Figure 12: Composites of T2m daily anomalies in November-March for observations (ERAI) and forecast (Model) for week 3 following the MJO phase 3. Dotted regions indicate statistical significance. Number in the upper right corner represents the pattern correlations between the two maps.

The Model's deficiencies are reflected by the weak anomaly correlation (0.32) between the Model and ERAI. Regionally, the Model misses the center of positive anomalies along the east coast of North America and Eurasia and the center of negative temperature anomalies over the North Pole.

This diagnostic helps identify if the forecast captures the specific temperature anomaly patterns in different regions of the NH, if the model predicts the sign reversal of T2m anomalies seen in observations for the opposite MJO phases, and the strength of the MJO impact on the surface air temperature.

#### 3. Summary, limitations and future development

The MJO-teleconnections diagnostics package is a Python tool that provides process level evaluation of the MJO and its teleconnections predicted by forecast models. The package is driven by a GUI that facilitates handling of large and complex datasets that are typically generated by forecast systems. The evaluation has flexibility to be conducted using an included dataset or user specified datasets. The diagnostics included in the package have been applied to peer reviewed studies and calculations are consistent with previous results. The list of diagnostics, in the order they appear on the GUI, includes:

- The STRIPES index for geopotential height at 500 hPa identifies regions where forecast systems capture/miss the
  impact of MJO across all its phases onto the extratropics large-scale circulation of the northern and southern
  hemispheres during boreal winter.
- The STRIPES index for precipitation identifies regions where extratropical precipitation in the forecast data is influenced or not by the MJO events present in the initial conditions of the forecasts during boreal winter.
- The Pattern CC and Relative amplitude for the PNA region (20°–80°N, 120°E–60°W) measures the forecast skill (pattern correlation) of the MJO teleconnections over the PNA region following the MJO convective activity located over the central Indian Ocean and eastern Maritime Continent (phases 2-3) and western central Pacific and central Pacific (phases 6-7). The relative amplitude provides a quantitative measure of the amplitude of the response in the forecast relative to verification.
- The Pattern CC and Relative amplitude for the Euro-Atlantic sector (20°-80°N, 60°W-90°E) provides the same information as the diagnostics for the PNA region except for the Euro-Atlantic sector.
- The Stratospheric Pathway identifies model deficiencies in forecasting the stratosphere-troposphere coupling mechanism driving the MJO teleconnections following the stratospheric pathway. The diagnostics evaluates the amplitude of meridionally averaged meridional heat flux in the middle troposphere (500 hPa) for the upward leg (weeks 2-3) and downward leg of the coupling (weeks 4-5) as well as the response of the polar vortex measured by the 100 hPa geopotential height.
- The Histogram of 10 hPa zonal wind provides information on the forecast system ability to predict the climatological distribution of the polar vortex winds. This diagnostic is a good indicator of the model limitations caused by the number of vertical levels and model top as model with a low-top struggle to tap into the soured of predictability arising from the stratospheric pathway (Stan et al. 2022).
- The MJO diagnostic provides the model skill in predicting the MJO amplitude and phase speed. The Hovmoller diagram of OLR and zonal wind at 850 hPa after MJO events in phases 2-3 are present in initial condition of forecasts provides an estimation of model's ability to predict the propagation of MJO activity across the Maritime Continent.
- The composites of T2m after MJO events in phases 3 and 4 present in the initial conditions of forecasts provides information on model's ability to tap into the source of predictability associated with the MJO.

The modular structure of the diagnostics allows for future expansion of the codebase to compute other diagnostics. For example, diagnostics for evaluation of biases in the models' mean state would provide a complete picture for the model evaluation. Other diagnostics such as the wave activity flux (WAF; Plumb 1985, Takaya and Nakamura 2001) and the stationary wavenumber on the Mercator projection (Hoskins and Ambrizzi 1993) are sometimes used to evaluate the Rossby wave propagation and the Rossby waveguides. These diagnostics will provide information on the model ability to predict the generation of Rossby waves with the observed propagation characteristics and model ability to predict unbiased mean states in the subtropics (e.g., subtropical westerly jet).

Further improvements in the regridding algorithm are warranted to reduce the calculation run time. The package could also be improved to allow direct model to model comparison when possible.

# Appendix A: Data pre-processing

Table A1: Variables accepted by the package for forecast data. The time variable must have units of days since the starting date of the forecast.

| Meteorological parameter             | Variable name in the data file              | Unit                                                             | Dimensions                   |
|--------------------------------------|---------------------------------------------|------------------------------------------------------------------|------------------------------|
| Geopotential at 500 hPa              | Any of: 'z',' Z',' gh', 'z500'              | Any of: 'm**2 s**2',<br>'m^2/s^2', 'm2/s2', 'm2s-2',<br>'m2 s-2' | (time#, latitude, longitude) |
| Zonal wind at 850 hPa                | Any of: 'u', 'U', 'uwnd', 'u850', 'uwnd850' | m/s                                                              | (time#, latitude, longitude) |
| Zonal wind at 200 hPa                | Any of: 'u', 'U', 'uwnd', 'u200', 'uwnd200' | m/s                                                              | (time, latitude, longitude)  |
| Zonal wind at 10 hPa                 | Any of: 'u', 'U', 'uwnd', 'u10', 'uwnd10'   | m/s                                                              | (time, latitude, longitude)  |
| Meridional wind at 850 hPa           | Any of: 'v', 'V', 'vwnd', 'v850', 'vwnd850' | m/s                                                              | (time#, latitude, longitude) |
| Meridional wind at 500 hPa           | Any of: 'v', 'V', 'vwnd', 'v500'            | m/s                                                              | (time, latitude, longitude)  |
| Outgoing Longwave<br>Radiation (OLR) | Any of: 'olr', 'ulwf'                       | Any of: 'w/m^2', 'w/m**2'                                        | (time, latitude, longitude)  |

| Precipitation rate     | Any of: 'prate', 'precipitationCal', 'precipitation', 'precip' | mm/day | (time, latitude, longitude) |
|------------------------|----------------------------------------------------------------|--------|-----------------------------|
| Temperature at 500 hPa | Any of: 'T', 't', 'temp', 't500'                               | K      | (time, latitude, longitude) |
| 2-meter Temperature    | Any of: 't2m', 'T2m', 'T', 'temp'                              | K      | (time, latitude, longitude) |

- <sup>#</sup>The extratropical cyclone activity diagnostics recommends using 6-hourly data. For those fields time should be replaced by
- forecast hour.

# 419 Code and data availability

- The package is available at https://doi.org/10.5281/zenodo.15002615 (Stan et al., 2025). Model data is available at
- https://registry.opendata.aws/noaa-ufs-s2s/, and ERAI data used as default verification by the package is available at
- https://drive.google.com/drive/folders/1wT51DRQhbXPAzVwvCWIkcojvWnCp7tgm?usp=sharing

#### 423 Author contributions

- All authors have contributed to this work. Conceptualization: CS. GUI development: SK and CS. Diagnostics development:
- CS, AMJ, JW, ZW, CZ, AS. Writing: All.

## 426 Competing interests

The authors have no competing interests.

# 428 Acknowledgements

This study was supported by the NOAA/OAR Weather Program Office through grant NA22OAR4590216.

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
