# Peer review of "A Python diagnostics package for evaluation of MJO-Teleconnections in S2S forecast systems"

_EGUsphere, 2025_

## Author Response (AR1)

**Reviewer 1**

The diagnostic and interface described have the potential to be an extremely useful package to diagnose MJO teleconnection performance in forecasting models that will be adopted by forecasting centers. Importantly, the diagnostics presented in this package are justified based on documentation in the peer-reviewed literature and physically sound. I commend the authors for the amount of work required to generate a package such as this, as it is important work and no small task.

We would like to thank the reviewer for the appreciation of our work and suggestions on how to improve the manuscript.

That said, the paper requires major revisions before acceptance, and many opportunities are missed to provide a much more compelling description and advertisement of this effort that will enhance uptake.

We addressed all concerns expressed by the reviewer and incorporated all recommended suggestions to the best of our abilities.

First, there are many strong individual contributions plugged into this paper, but the paper doesn't flow well, doesn't have a unified voice, and is disjointed. One or two authors need to take the initiative to provide such a cohesive narrative. In places, the manuscript reads like a collection of parts, and more effort is needed to make the whole greater than the sum of parts.

We revised the flow of the manuscript and improved the narrative.

Second, results from a model are presented alongside observations analysis, but little if any insight is ever provided into the nature of deficiencies in the model and why the particular diagnostic being examined provides insight into model performance. The model results just seem like extra figures that are not discussed.

We expanded the description of model errors identified by the diagnostics. We would also like to note that a model evaluation is also intended to determine how well the model is predicting the MJO-Teleconnections, not only to identify the model limitations.

Third, the summary section is disappointingly terse and fragmented, and misses an opportunity for a more thorough and enlightening discussion of limitations, challenges, and where the package will develop in the future. This section reads like an afterthought that was written very quickly.

We also expanded the summary section. In terms of the package development, we don't anticipate expanding it into a larger application. By keeping it smaller, it remains simple, it can be modified by advanced users and it can easily be incorporated into operational forecast verification systems, e.g, METPlus. For this reason, the package can also be used without the GUI.

Lastly, more details on format of data inputs might be provided. Further comments are listed below.

Mode details have been included.

**Comments:**

• Lines 52-55. This description of the mechanism through which the MJO produces a stationary Rossby wave teleconnection is terse and might be expanded, given the emphasis of the paper.

The description of Rossby wave generation has been added:

L52-56: The Rossby waves form in response to perturbations induced by moist diabatic processes associated with tropical convection (Teng and Branstator 2019), which oftentimes aggregates into the MJO. The waves are generated by the so called Rossby wave sources (Sardeshmukh and Hoskins, 1988) or basic state vorticity gradients. The MJO heating leads to horizontal divergence of the wind in the upper troposphere and changes in the rotational wind. The wave activity emanating from the source propagates eastward and poleward.

• In general, I find the writing in the introduction to be choppy and not flow well, and so could use some work to improve the narrative.

We expanded the Introduction and explained the objective of the paper.

L73-75: The response of the extratropics to MJO forcing establishes in a two-week time scale (Jin and Hoskins, 1995) and depends on the MJO phases described by the location of active and suppressed convection. Stan et al. 2017 provides a review of mechanisms underpinning the tropical-extratropical teleconnections.

L79-81: The scientific basis of diagnostics included in the package have been documented in literature. The diagnostics have been applied to the forecast systems in the S2S database (Stan et al. 2022) and the prototypes of the NOAA UFS global coupled model (Zheng et al. 2024, Garfinkel et al. 2024, Wang et al. 2025).

L83-88: The objective of the paper is to guide users in how to apply the package to their forecast data, understand the strength and weaknesses of a forecast system in predicting the mechanisms driving the MJO teleconnections compared to their observed characteristics, and to provide a limited deterministic evaluation of the forecast skill. Additionally, the package provides a tool for evaluation of the MJO forecast skill. Due to the delayed response of the extra-tropics to MJO forcing, evaluation of MJO teleconnections by forecast systems is conducted with respect to the presence of MJO events in initial conditions, which allows the usage of reanalysis/observation based products for event description.

• Many challenges arise when developing a diagnostic package for community use, for example:

Neelin, J. D., J. P. Krasting, A. Radhakrishnan, J. Liptak, T. Jackson, Y. Ming, W. Dong, A. Gettelman, D. R. Coleman, E. D. Maloney, A. A. Wing, Y.-H. Kuo, F. Ahmed, P. Ullrich, C. M. Bitz, R. B. Neale, A. Ordonez, and Elizabeth A. Maroon, 2023: Process-oriented diagnostics: principles, practice, community development and common standards. *Bull. Amer. Meteor. Soc.*, **104**, E1452–E1468.

It would be worthwhile to acknowledge this and also discuss how this diagnostic package overcomes or hopes to overcome such challenges. A lot can go wrong with implementation of a package like this that seems simple on its face, but is complicated to implement in detail. As such, it would be good for this group to interface with the NOAA Model Diagnostics Task Force to share experiences and lessons.

The MDTF and other diagnostics packages have been acknowledged in the introduction.

L91-100: The basic concept of the diagnostics package is similar to other community-contributed metrics packages such as the PCMDI Metric Package (PMP; Gleckler et al. 2008, 2016; Lee et al. 2024), the Toolkit for Extreme Climate Analysis (TECA), the international Land Model Benchmarking Tool (ILAMB; Collier et al. 2018), the International Ocean Model Benchmarking (IOMB) package operating under the umbrella of the Coordinated Model Evaluation Capabilities (CMEC), the Processoriented diagnostics (PODs) coordinated by the Model Diagnostics Task Force (MDTF; Neelin et al. 2023), the Climate Variability and Diagnostics Package (CDVP; Phillips et al. 2014, Maher et al. 2024), the Atmosphere Model Working Group (AMWG) Diagnostics Framework (ADF), and others. The major difference between these diagnostics packages and the MJO-Teleconnections diagnostics package is tailored for S2S forecast data, which typically extend to no more than 46 days. The other packages apply to multi-century climate simulations.

• Lines 120-131. The devil may be in the details and a lot can go wrong in this standardization of data. Some discussion of lessons learned when different models are entrained, as well as discussion of the importance of standardization (e.g. Neelin et al. 2023) might be helpful.

Additional details regarding the description of coordinates have been included:

L157-159: the ERAI grid (512 longitude grid points ordered from 0o to 360o, 256 latitude grid points ordered from 90N to 90S) and IMERG grid (480 longitude grid points ordered from 0o to 360o, 241 latitude grid points ordered from 90S to 90N), respectively. The stored direction of forecast data's latitude can be either decreasing or increasing.

• There is a lot unclear about the details, including what temporal resolution forecast data can be used, what number of forecast lags can be used, etc.

The temporal resolution required by the diagnostics it was specified at line 136 in the initial submission and line xx in the revised manuscript and in Table A1 in the Appendix A. The S2S forecast systems typically consist of maximum 46 days, because there is no skill at longer lags. The MJO teleconnections are typically relevant for weeks 3-5. A clarification was added to address the required forecast length:

L170: The forecast data must consist of a minimum of 35 days.

• Figures 3 and 4. No interpretation of what these figures are telling us about model performance is provided. For example, is the bottom row of Figure 3 concerning, good, or somewhere in between regarding model performance? Providing evidence that these diagnostics are insightful will help uptake of the package.

The general description of the STRIPES index is now separated into two parts: the first part is a direct description of Fig. 3 and the second part retains the general description.

L188-194: The STRIPES index for Z500 shows that in ERAI the strongest impact of MJO manifests over regions along the Atlantic and Pacific storm tracks and Europe. In the Southern Hemisphere, the response to MJO forcing is a zonally elongated belt around 60S. The forecast captures the approximate location of the response centers in both hemispheres. However, the magnitude of the response is weaker in the Northern Hemisphere and stronger in the Southern Hemisphere as shown by the difference plot. The STRIPES index applied to IMERG shows that the response of extratropical

precipitation to MJO forcing is localized over the same regions as the circulation response. The Model forecasts a weaker than observed amplitude of the response in both hemispheres.

The interpretation of Fig. 4 has also been expanded:

L201-2016: A comparison of the STRIPES index for the two periods shows that calculation based on the shorter period yields in maximum values of the STRIPES index larger than in the calculation based on a longer period.

• Figure 6. Again, a bit more insight into model deficiencies that this diagnostic provides would be helpful.

Description of Fig. 5 and 6 have been expanded:

L219-227: Figure 5 shows the metrics for the PNA region when MJO events in phases 2-3 and 6-7 are present in the initial conditions of the forecasts. In this example, the prediction of MJO teleconnection pattern is skilful (pattern CC > 0.6) up to two weeks regardless of the MJO phase. In the first week, the amplitude of MJO teleconnection in the forecast data is close to ERAI (relative amplitude ~ 1). As the lead time increases, the Model alternates between underestimating (relative amplitude < 1) and overestimating (relative amplitude > 1) the amplitude of the MJO teleconnections to the PNA region for both MJO phases. At shorter leads (weeks 2-3) the amplitude is largely underestimated following the MJO phases 6-7 whereas at longer leads (weeks 3-4) the overestimation of amplitude is more pronounced for the MJO phases 2-3.

L236-242: Figure 6 shows that in ERAI, the Z500 composites after the MJO phases 2-3 resemble the negative PNA pattern and the composites after the MJO phases 6-7 resemble the positive PNA pattern. The composites based on the forecast data capture the Z500 anomaly pattern well in the first two weeks of the forecast after both the MJO phases 2-3 and 6-7 and the accuracy decreases at longer leads. One particular aspect to notice is the model failure to capture the transition of Z500 anomalies associated with the MJO phases 2-3 to the opposite MJO phases (6-7) occurring in week 3 in ERAI. The pattern of the MJO teleconnections in the forecast tends to agree better with ERAI over the North Pacific than over North America indicating some model deficiencies in predicting the patterns over the land than over the ocean.

• Figure 8. Again here, what insight is provided into this particular model by examining the right column of this figure? It seems unnecessary to include panels from a model in this paper if they are not discussed and interpreted.

A description of model's deficiencies has been added in the revised version of the manuscript:

L286-289: In this case, the Model is not able to maintain the strength of the positive heat flux associated with the MJO phase 5 beyond week 2. In fact, the Model reverses the sign of the heat flux anomaly at longer forecast leads, suggesting an opposite response of the polar vortex, which is confirmed by the 100 hPa geopotential height anomalies averaged over the polar cap seen in weeks 3-5 following the MJO phase 5.

• Section: 2.2.6 MJO. This section provides information on basic MJO performance, and seems out of order relative to the rest of the plots that discuss teleconnections. This section might be moved to be the first diagnostic discussed. This relates to my comment above about the manuscript not flowing well.

It would be nice to have an MJO and MJO-Teleconnections diagnostics package, but these diagnostics are focused on teleconnections only. As explained above, the MJO diagnostic is a bonus diagnostic because the MJO teleconnections do not depend on the model ability to forecast the MJO. The RMM index used in the calculation of teleconnections is not based on forecast data. The MJO diagnostics are simply metrics that do not reveal much about processes driving the MJO.

• Summary, limitations and future development. I was expecting a more thorough and earnest discussion of the limitations of the manuscript and future developments of the package, but this section is disappointingly terse and cursory, with fragments of ideas. The ideas in this section need to be expanded.

The section has been updated:

L385-410: The list of diagnostics, in the order they appear on the GUI, includes:

- The STRIPES index for geopotential height at 500 hPa. identifies regions where forecast systems capture/miss the impact of MJO across all phases onto the extratropics large-scale circulation of the northern and southern hemispheres during boreal winter.
- The STRIPES index for precipitation identifies regions where extra-tropical precipitation in the forecast data is influenced or not by the MJO events present in the initial conditions of the forecasts during boreal winter.
- The Pattern CC and Relative amplitude for the PNA region (20°–80°N, 120°E–60°W) measures the forecast skill (pattern correlation) of the MJO teleconnections over the PNA region following the MJO convective activity located over the central Indian Ocean and eastern Maritime Continent (phases 2-3) and western central Pacific and central Pacific (phases 6-7). The relative amplitude provides a quantitative measure of the amplitude of the response in the forecast relative to verification.

- The Pattern CC and Relative amplitude for the Euro-Atlantic sector (20°–80°N, 60°W–90°E) provides the same information as the diagnostics for the PNA region except for the Euro-Atlantic sector.
- The Stratospheric Pathway identifies model deficiencies in forecasting the stratosphere-troposphere coupling mechanism driving the MJO teleconnections following the stratospheric pathway. The diagnostics evaluates the amplitude of meridionally averaged meridional heat flux in the middle troposphere (500 hPa) for the upward leg (weeks 2-3) and downward leg of the coupling (weeks 4-5) as well as the response of the polar vortex measured by the 100 hPa geopotential height.
- The Histogram of 10 hPa zonal wind provides information on the forecast system ability to predict the climatological distribution of the polar vortex winds. This diagnostic is a good indicator of the model limitations caused by the number of vertical levels and model top as model with a low-top struggle to tap into the soured of predictability arising from the stratospheric pathway (Stan et al. 2022).
- The MJO diagnostic provides the model skill in predicting the MJO amplitude and phase speed. The Hovmoller diagram of OLR and zonal wind at 850 hPa after MJO events in phases 2-3 are present in initial condition of forecasts provides an estimation of model's ability to predict the propagation of MJO activity across the Maritime Continent.
- The composites of T2m after MJO events in phases 3 and 4 present in the initial conditions of forecasts provides information on model's ability to tap into the source of predictability associated with the MJO.

L414-418: Other diagnostics such as the wave activity flux (WAF; Plumb 1985, Takaya and Nakamura 2001) and the stationary wavenumber on the Mercator projection (Hoskins and Ambrizzi 1993) are sometimes used to evaluate the Rossby wave propagation and the Rossby waveguides. These diagnostics will provide information on the model ability to predict the generation of Rossby waves with the observed propagation characteristics and model ability to predict unbiased mean states in the subtropics (e.g., subtropical westerly jet).

L419-420: The package could also be improved to allow direct model to model comparison when possible.

**Reviewer 2**

This has the potential to be a very useful package for the model evaluation community and a detailed description of the capabilities and applications will be paramount to the users of this package.

We thank the reviewer for appreciating the potential of the presented diagnostic and suggestions on how to improve the manuscript.

Overall the manuscript and in particular the summary at the end is very brief and short on details. I would expect the summary to include a discussion of how this package will assist users in evaluating model skill. Ideally, this section should also include a paragraph on how these diagnostics have helped improve models in the past and how they were applied in those cases. Another aspect that is missing from the manuscript in present form how the different diagnostics discussed in the manuscript fit together to give a comprehensive view of MJO skill and/ or what aspects are still missing and are not touched on by these metrics. For example, one useful application that is currently not available is the capability to compare verification and multiple sets of models, this should be highlighted in the summary (I think it is mentioned briefly in section 2.2).

The summary has been expanded.

L385-410: The list of diagnostics, in the order they appear on the GUI, includes:

- The STRIPES index for geopotential height at 500 hPa identifies regions where forecast systems capture/miss the impact of MJO across all phases onto the extratropics large-scale circulation of the northern and southern hemispheres during boreal winter.
- The STRIPES index for precipitation identifies regions where extra-tropical precipitation in the forecast data is influenced or not by the MJO events present in the initial conditions of the forecasts during boreal winter.
- The Pattern CC and Relative amplitude for the PNA region (20°–80°N, 120°E–60°W) measures the forecast skill (pattern correlation) of the MJO teleconnections over the PNA region following the MJO convective activity located over the central Indian Ocean and eastern Maritime Continent (phases 2-3) and western central Pacific and central Pacific (phases 6-7). The relative amplitude provides a quantitative measure of the amplitude of the response in the forecast relative to verification.
- The Pattern CC and Relative amplitude for the Euro-Atlantic sector (20°–80°N, 60°W–90°E) provides the same information as the diagnostics for the PNA region except for the Euro-Atlantic sector.
- The Stratospheric Pathway identifies model deficiencies in forecasting the stratosphere-troposphere coupling mechanism driving the MJO teleconnections following the stratospheric pathway. The diagnostics evaluates the amplitude of meridionally averaged meridional heat flux in the middle troposphere (500 hPa) for the upward leg (weeks 2-3) and downward leg of the coupling (weeks 4-5) as well as the response of the polar vortex measured by the 100 hPa geopotential height.

- The Histogram of 10 hPa zonal wind provides information on the forecast system ability to predict the climatological distribution of the polar vortex winds. This diagnostic is a good indicator of the model limitations caused by the number of vertical levels and model top as model with a low-top struggle to tap into the soured of predictability arising from the stratospheric pathway (Stan et al. 2022).
- The MJO diagnostic provides the model skill in predicting the MJO amplitude and phase speed. The Hovmoller diagram of OLR and zonal wind at 850 hPa after MJO events in phases 2-3 are present in initial condition of forecasts provides an estimation of model's ability to predict the propagation of MJO activity across the Maritime Continent.
- The composites of T2m after MJO events in phases 3 and 4 present in the initial conditions of forecasts provides information on model's ability to tap into the source of predictability associated with the MJO.

L414-418: Other diagnostics such as the wave activity flux (WAF; Plumb 1985, Takaya and Nakamura 2001) and the stationary wavenumber on the Mercator projection (Hoskins and Ambrizzi 1993) are sometimes used to evaluate the Rossby wave propagation and the Rossby waveguides. These diagnostics will provide information on the model ability to predict the generation of Rossby waves with the observed propagation characteristics and model ability to predict unbiased mean states in the subtropics (e.g., subtropical westerly jet).

L419-420: The package could also be improved to allow direct model to model comparison when possible.

It would be nice to have an MJO and MJO-Teleconnections diagnostic package, but these diagnostics is focused on MJO-Teleconnections. The package includes two metrics for the MJO skill. The MJO diagnostic is a bonus diagnostic because the MJO teleconnections do not depend on the model ability to forecast the MJO. The RMM index used in the calculation of teleconnections is not based on forecast data. The MJO diagnostics are simply metrics that do not reveal much about processes driving the MJO.

Page 2, lines 44-50: I don't think this is necessary in the introduction, especially since the author's don't refer to the phases again in the introduction. It feels out of place and should maybe go into a separate section on the MJO and which aspects are relevant for prediction skill and which of those to diagnostics aim to address. This detailed description feels out of place here. This is also one of only a few places in the manuscript that go into great detail on the MJO, in contrast most of the manuscript does not discuss MJO dynamics or thermodynamics in detail. This feels out of balance. The authors could consider adding a section describing the relevant aspects of the MJO in detail and then refer back to that when discussing each diagnostic in detail to highlight which aspects of the MJO each diagnostic addresses.

The only aspects of the MJO that are relevant for the MJO teleconnections is the location of the MJO convection also known as the MJO phases. Every diagnostic invokes some of the MJO phases. The MJO dynamics and thermodynamics are not relevant for the MJO teleconnections because only the MJO events present in the initial conditions are considered by the diagnostics. This is the case because the MJO teleconnections to the extratropics establishes in about 2 weeks following certain phases of the MJO. A clarification has been added to the manuscript:

L86-88: Due to the delayed response of the extra-tropics to MJO forcing, evaluation of MJO teleconnections by forecast systems is conducted with respect to the presence of MJO events in initial conditions, which allows the usage of reanalysis/observation based products for event description.

2. Showing the comparison to the model is appreciated. It would be useful to have a short discussion on why the exact model is not specified. I assume it is because the authors don't want to focus on the model evaluation, but rather the package capabilities. Even so, I think it is still important, when showing each diagnostic and comparison between ERAI/IMERG and model, to include how the application of the diagnostic helps inform forecasts and/ or users. This is currently not done. Given that the main point of diagnostics is to improve models and characterize their skill compared to other models these sections need more details on how this is achieved.

A description of limitations in the Model used as an example is now added to each diagnostic:

L188-194: The STRIPES index for Z500 in the NH shows that in ERAI the strongest impact of MJO manifests over regions along the Atlantic and Pacific storm tracks and Europe. In the Southern Hemisphere (SH), the response to MJO forcing is a zonally elongated belt around 60S. The forecast captures the approximate location of the response centers in both hemispheres. However, the magnitude of the response is weaker in the NH and stronger in the SH as shown by the difference plot. The STRIPES index applied to IMERG shows that the response of extratropical precipitation to MJO forcing is localized over the same regions as the circulation response. The Model forecasts a weaker than observed amplitude of the response in both hemispheres.

L220-227: In this example, the prediction of MJO teleconnection pattern is skillful (pattern CC > 0.6) up to two weeks regardless of the MJO phase. In the first week, the amplitude of MJO teleconnection in the forecast data is close to ERAI (relative amplitude ~ 1). As the lead time increases, the Model alternates between underestimating (relative amplitude < 1) and overestimating (relative amplitude > 1) the amplitude of the MJO teleconnections to the PNA region for both MJO phases. At shorter leads (weeks 2-3) the amplitude is largely underestimated following the MJO phases 6-7 whereas at longer leads (weeks 3-4) the overestimation of amplitude is more pronounced for the MJO phases 2-3

L236-242: Figure 6 shows that in ERAI, the Z500 composites after the MJO phases 2-3 resemble the negative PNA pattern and the composites after the MJO phases 6-7 resemble the positive PNA pattern. The composites based on the forecast data capture the Z500 anomaly pattern well in the first two weeks of the forecast after both the MJO phases 2-3 and 6-7 and the accuracy decreases at longer leads. One particular aspect to notice is the model failure to capture the transition of Z500 anomalies associated with the MJO phases 2-3 to the opposite MJO phases (6-7) occurring in week 3 in ERAI. The pattern of the MJO teleconnections in the forecast tends to agree better with ERAI over the North Pacific than over North America indicating some model deficiencies in predicting the patterns over the land than over the ocean.

L286-289: In this case, the Model is not able to maintain the strength of the positive heat flux associated with the MJO phase 5 beyond week 2. In fact, the Model reverses the sign of the heat flux anomaly at longer forecast leads, suggesting an opposite response of the polar vortex, which is confirmed by the 100 hPa geopotential height anomalies averaged over the polar cap seen in weeks 3-5 following the MJO phase 5.

L350-352: The AERR and PERR metrics show that the Model predicts MJO events with weaker amplitude and faster phase speed than in ERAI. The ACC shows that the forecast skill drops below 0.5 after ~27 days at about the same time when RMSE = 1.4. The Hovmoller diagram shows that in the model convective activity takes a longer time to propagate across the Maritime continent.

L379-381:The Model's deficiencies are reflected by the weak anomaly corelation (0.32) between the Model and ERAI. Regionally, the Model misses the center of positive anomalies along the east coast of North America and Eurasia and the center of negative temperature anomalies over the North Pole.